# Synthesis of Deuterium-Labeled Vitamin D Metabolites as Internal Standards for LC-MS Analysis

**DOI:** 10.3390/molecules27082427

**Published:** 2022-04-09

**Authors:** Akiko Nagata, Kazuto Iijima, Ryota Sakamoto, Yuka Mizumoto, Miho Iwaki, Masaki Takiwaki, Yoshikuni Kikutani, Seketsu Fukuzawa, Minami Odagi, Masayuki Tera, Kazuo Nagasawa

**Affiliations:** 1Department of Biotechnology and Life Science, Tokyo University of Agriculture and Technology, Koganei, Tokyo 184-8588, Japan; s193819x@st.go.tuat.ac.jp (A.N.); s206189q@st.go.tuat.ac.jp (K.I.); s218969x@st.go.tuat.ac.jp (R.S.); s200343y@st.go.tuat.ac.jp (Y.M.); s218363r@st.go.tuat.ac.jp (M.I.); odagi@cc.tuat.ac.jp (M.O.); tera@go.tuat.ac.jp (M.T.); 2Medical Equipment Business Operations, Management Strategy Planning Division, JEOL Ltd., Akishima, Tokyo 196-8558, Japan; mtakiwak@jeol.co.jp (M.T.); ykikutan@jeol.co.jp (Y.K.); sfukuzaw@jeol.co.jp (S.F.)

**Keywords:** vitamin D, deuterium labeling, liquid-chromatography tandem mass spectrometry, measurement of vitamin D metabolites in blood

## Abstract

Blood levels of the vitamin D_3_ (D_3_) metabolites 25-hydroxyvitamin D_3_ (25(OH)D_3_), 24*R*,25-dihydroxyvitamin D_3_, and 1α,25-dihydroxyvitamin D_3_ (1,25(OH)_2_D_3_) are recognized indicators for the diagnosis of bone metabolism-related diseases, D_3_ deficiency-related diseases, and hypercalcemia, and are generally measured by liquid-chromatography tandem mass spectrometry (LC-MS/MS) using an isotope dilution method. However, other D_3_ metabolites, such as 20-hydroxyvitamin D_3_ and lactone D_3_, also show interesting biological activities and stable isotope-labeled derivatives are required for LC-MS/MS analysis of their concentrations in serum. Here, we describe a versatile synthesis of deuterium-labeled D_3_ metabolites using A-ring synthons containing three deuterium atoms. Deuterium-labeled 25(OH)D_3_ (**2**), 25(OH)D_3_-23,26-lactone (**6**), and 1,25(OH)_2_D_3_-23,26-lactone (**7**) were synthesized, and successfully applied as internal standards for the measurement of these compounds in pooled human serum. This is the first quantification of 1,25(OH)_2_D_3_-23,26-lactone (**7**) in human serum.

## 1. Introduction

Vitamin D_3_ (D_3_) (**1**) is metabolized by members of the cytochrome P450 (CYP) family to generate more than 50 compounds in vivo. Among them, 25-hydroxyvitamin D_3_ (25(OH)D_3_) (**2**) is generated from D_3_ (**1**) by CYP2R1 and/or CYP27A1-mediated hydroxylation at C25 in the liver, and the resulting 25(OH)D_3_ (**2**) is further metabolized to the active form of D_3_, 1α,25-dihydroxyvitamin D_3_ (1,25(OH)_2_D_3_) (**3**), by CYP27B1-mediated oxidation at C1α in the kidneys (Figure 1). 1,25(OH)_2_D_3_ (**3**) plays a key role in the regulation of bone metabolism in vivo [1]. In normal conditions, production of **3** from **2** is strictly controlled by the concentrations of calcium and parathyroid hormone (PTH) in the blood, and thus the concentration of **3** in the blood is a useful indicator of functional status [2] and is helpful in the diagnosis of diseases such as hypercalcemia, hyperphosphatemia, rickets, and bone metabolism-related diseases [3,4]. The concentration of **2** in the blood is also useful as an indicator for the diagnosis of various vitamin D deficiency-related diseases [3,4]. Recently, various D_3_ metabolites, mostly oxidized at the D-ring side chain, have also been found to show biological activities. For example, 20*S*,25(OH)_2_D_3_, which is generated by CYP11A1, inhibits the growth of keratinocytes, leukemia cells, and melanoma cells [5,6,7,8,9], while 24*R*,25(OH)_2_D_3_ (**4**), produced by CYP24A1, shows inhibitory activity against various cancer cell lines [10,11]. Further, 25(OH)D_3_-23,26 lactone (**6**) and 1,25(OH)_2_D_3_-23,26 lactone (**7**), which are thought to be final metabolites of D_3_, show antagonistic activity towards 1,25(OH)_2_D_3_ [12,13,14], thereby, inhibiting bone formation and resorption. Recently, compound **7** was reported to inhibit fatty acid oxidation [15]. Thus, there is a need to measure the blood levels of these metabolites.

The concentrations of metabolites **2** and **3** in blood have been measured for clinical purposes by radioimmunoassay (RIA) or chemiluminescent enzyme immunoassay (CLEIA) [16]. They, however, have disadvantages such as the need to handle radioactive materials, and insufficient discrimination of vitamin D metabolites by antibodies [17]. More recently, a liquid-chromatography tandem mass spectrometry (LC-MS/MS) method has been developed to determine the concentration of multiple vitamin D metabolites simultaneously in blood [18]. However, LC-MS/MS-based measurement also has some problems, such as the low ionization efficiency of vitamin D derivatives and interference by contaminants including multiple D_3_ metabolites in the blood. To address these issues, several approaches have been investigated. Cookson-type reagents have been developed to improve the ionization efficiency of D_3_ metabolites, affording high sensitivity even at low abundance [19,20]. The isotope dilution method has also been applied to avoid interference from contaminants in the blood. This method requires a stable isotope-labeled compound as an internal standard, and so far, deuterium-labeled 25(OH)_2_D_3_ (**2**), 1,25(OH)_2_D_3_ (**3**), and 24*R*,25(OH)_2_D_3_ (**4**), in which deuterium is introduced at C26, C27, C6, and C19, have been synthesized (Figure 2) [21,22,23,24,25].

In the synthesis of the deuterium-labeled metabolites **2**–**4**-*d*_6_, deuterium was introduced into the side chain at C26 and C27 by reacting esters **8** with deuterated Grignard reagent, CD_3_MgBr (Figure 2B) [21,22]. On the other hand, **2**–**3**-*d*_3_ were synthesized by reacting SO_2_ adducts of cyclic compounds **9** derived from D_3_ with deuterium oxide (D_2_O) [23,24,25]. In both strategies, the range of metabolites that can be synthesized is limited due to the restrictions imposed by the use of steroid precursors. Therefore, a more versatile approach is required. Convergent strategies, with coupling between CD-ring and A-ring moieties, have been widely applied for the synthesis of D_3_ derivatives [26,27]. Since the CD-ring structures of the metabolites are diverse, whereas the A-ring structures are relatively constant, we considered that deuterium-labeled A-ring synthons would be suitable for the preparation of a variety of deuterium-labeled D_3_ metabolites (Figure 2D). In addition, labeling in the A-ring has an advantage in metabolism studies because the side chains of the D3 are well known to be enzymatically metabolized easily. In this study, we have developed a synthesis of deuterium-labeled A-ring precursors **13***-d*_3_ and **16***-d*_3_ incorporating three deuterium atoms. These precursors were coupled with CD-ring moieties **17** and **18** to afford deuterium-labeled 25(OH)D_3_*-d*_3_ (**2***-d*_3_) and vitamin D lactones 25(OH)D_3_-23,26-lactone-*d*_3_ (**6**-*d*_3_) and 1,25(OH)_2_D_3_-23,26-lactone-*d*_3_ (**7**-*d*_3_). We also confirmed that the concentrations of **2**, **6**, and **7** in human serum could be measured by LC-MS/MS using the corresponding deuterium-labeled compounds as the internal standards (IS) (see Appendix A).

## 2. Results

We employed a convergent strategy using the palladium-catalyzed coupling reaction of enyne-type deuterium-labeled A-ring precursors **13***-d*_3_ and **16***-d*_3_ with bromoolefins **17** and **18** as the CD-ring moieties. The deuterium atoms in **13***-d_3_* and **16***-d*_3_ were introduced by the H/D exchange at the a-position of the alcohol, as reported by Sajiki et al. [28]. Our synthesis of deuterium-labeled enyne **13** commenced with the H/D exchange reaction of alcohol **10**, which was obtained from L-(-)-malic acid in 4 steps (Figure 1) [29]. The alcohol **10** was subjected to the H/D exchange reaction with a catalytic amount of Ru/C in D_2_O at 80 °C under an H_2_ atmosphere to afford **10***-d*_3_ deuterium-labeled at C3 and C4 in a 96% yield with over 93% deuteride content [28]. In this reaction, the stereochemistry at C3 was isomerized (4:1 ratio of α-**10a** and β-**10b**). The deuterium-labeled alcohol **10** (enantiomeric mixture) was converted into alkyne **11** by tosylation of the primary alcohol followed by epoxidation with NaH and reaction with TMS-acetylene (22% yield from **10***-d*_3_). The hydroxyl group in alkyne **11** was protected with TBS ether, followed by deprotection of the TMS and pivaloyl groups with NaOMe in MeOH to give the alcohol **12** in an 86% yield from **11**. Enyne **13** was obtained in a 51% yield from **12** via 4 steps, (i) tosylation of the primary alcohol; (ii) cyanation with NaCN; (iii) reduction of the nitrile group with DIBAL-H to aldehyde; and (iv) a Wittig reaction with Ph_3_PCH_3_I and NaHMDS. It was confirmed by ^1^H-NMR that the deuteration rate did not decrease in these reaction steps [30].

Next, the deuterium-labeled enyne **16** bearing a hydroxyl group at C1α was synthesized (Figure 2). The alcohol moiety in **12** was oxidized with an IBX and the resulting aldehyde was reacted with a HWE Wittig reagent to give an unsaturated ester, whose ester group was reduced with a DIBAL-H to give allyl alcohol **14** in a 70% yield from **12** [31,32]. The allyl alcohol **14** was subjected to a Sharpless asymmetric epoxidation with a TBHP in the presence of Ti(OiPr)_4_ and L-(+)-DET [33], and the resulting epoxy alcohol was subjected to iodination with iodine and triphenylphosphine followed by treatment with zinc to give a secondary alcohol **15** in a 79% yield (3 steps) [31,32]. The diastereomer ratio at C1 in **15** was 10:1, and the undesired C1β diastereomer was removed by kinetic resolution, using acylation with isopropyl acid anhydride in the presence of (2*S*,3*R*)-HyperBTM [34], to give (-)-**15** in an 82% yield as a single diastereomer. The undesired diastereomer at C3 was also removed via silica gel column purification. The deuterium-labeled enyne **16**, in which the secondary alcohol was protected as the TBS ether, was obtained in an 82% yield.

The palladium-catalyzed coupling reaction of **13***-d*_3_ and bromoolefin **17** followed by deprotection of the silyl groups provided 25(OH)D_3_*-d*_3_ (**2**-*d*_3_) in a 36% yield [35]. Next, 25(OH)D_3_-23,26-lactone*-d_3_* (**6***-d*_3_) and 1,25(OH)_2_D_3_-23,26-lactone*-d*_3_ (**7***-d*_3_) were similarly synthesized by reacting bromoolefin **18** and enynes **13***-d*_3_ and **16***-d*_3_, respectively [36]. In the synthesis of **2***-d*_3_ and **6***-d*_3_, the undesired diastereomers at C3α were separated by an HPLC (Figure 3).

### 2.1. Derivatization of **2**, **6**, **7** for LC-MS/MS, and Preparation of Calibration Curves

With the deuterium-labeled D_3_ metabolites of **2***-d*_3_, **6***-d*_3_, and **7***-d*_3_ in hand, we next examined the quantitative analysis of the three D_3_ metabolites in pooled human serum by LC-MS/MS. First, we confirmed that our deuterium-labeled D_3_ metabolites were suitable as the internal standards for the isotope dilution method in an LC-MS/MS analysis. As described above, D_3_ and its metabolites have low ionization efficiency in an LC-MS/MS, and derivatization is necessary to improve the ionization efficiency. Thus, the D_3_ metabolites **2**, **6**, and **7**, as well as **2***-d*_3_, **6***-d*_3_, and **7**-*d*_3_, were derivatized with a recently developed reagent DAP-PA (4-(4′-dimethylaminophenyl)-1,2,4-triazoline-3,5-dione-phenyl anthracene) [20], and the ion peaks of the DAP adducts were detected by selective reaction monitoring (SRM) under the LC-MS/MS conditions shown in Table 1 (Figure 3).

In the case of the DAP-adducts of **2** and **2***-d*_3_ (Figure 3A,B), we observed identical ion peaks at the retention time of 5.60 min (abbreviated as *t*_R_: 5.60 min). Similarly, **6** and **6***-d*_3_ showed the same *t*_R_ of 3.50 min, and **7** and **7**-*d*_3_ showed the same *t*_R_ of 2.15 min, indicating that the deuterium-labeled compounds are suitable as internal standards for the isotope dilution method. We also observed small peaks at the retention times of 5.20 min (Figure 3A,B), 2.65 min (Figure 3C,D), and 2.37 min (Figure 3E,F) for **2**/**2***-d*_3_, **6**/**6***-d*_3_, and **7**/**7***-d*_3_, respectively. These peaks are due to the epimers at C6 of the DAP adducts, because DAP-PA reacts from both the α- and β-faces.

Next, the calibration curves were prepared as follows (Figure 4). A total of 100 μL of each one of the calibrator solutions was mixed with 200 μL of the internal standard solution and evaporated to dryness. After derivatization with DAP-PA, an LC-MS/MS analysis of the unlabeled and labeled DAP-adducts was performed, and the calibration curves were prepared by plotting the concentration of unlabeled DAP-adduct against the ion peak area ratio of unlabeled versus labeled DAP-adduct. All of the calibration curves showed good linearity.

### 2.2. Quantification of the D_3_ Derivatives in Human Serum

The levels of **2**, **6**, and **7** in pooled human serum were quantified by the LC-MS/MS using the isotope dilution method with the constructed calibration curves. The serum was pretreated as follows. An aliquot of serum (100 μL) was mixed with the internal standards solution (200 μL). Each sample was loaded onto a supported liquid extraction column (ISOLUTE SLE+ 300 μL sample Volume, Biotage, Uppsala, Sweden) and eluted three times with 600 mL hexane/ethyl acetate (1/1, *v*/*v*) using a PRESSURE+48 positive pressure manifold (Biotage, Uppsala, Sweden). The combined eluates were evaporated to dryness in a centrifugal evaporator. The ion peaks of the metabolites matched well with those of the corresponding internal standards in the pretreated samples (Figure 5). 

The concentrations of **2**, **6**, and **7** in human serum were calculated to be 5.1 ng/mL, 38.3 pg/mL, and 8.9 pg/mL, respectively, based on the area ratios of the detected peaks. The concentrations of **2** and **6** were in agreement with previously reported values [37], while this is the first quantification of 1α-lactone **7** in human serum.

## 3. Conclusions

We synthesized deuterium-labeled A-ring-*d_3_* synthons **13** and **16** and utilized them for the convergent synthesis of deuterium-labeled D_3_ derivatives 25(OH)D_3_ (**2**), 25(OH)D_3_-23, 26-lactone (**6**), and 1,25(OH)_2_D_3_-23, 26-lactone (**7**). These deuterium-labeled D_3_ metabolites were successfully applied as internal standards for the quantification of the metabolites in pooled human serum by LC-MS/MS using the isotope dilution method. This is the first quantification of 1,25(OH)_2_D_3_-23, 26-lactone (**7**) in human serum.

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
