# Peer review of "Synthesis of Deuterium-Labeled Vitamin D Metabolites as Internal Standards for LC-MS Analysis"

_molecules, 2022, doi:10.3390/molecules27082427_

Round 1
Reviewer 1 Report
The authors provide an interesting synthetic approach to deuterium-labeled vitamin D metabolites, that are multifold labeled in the A-ring of the vitamin D skeleton, to apply as internal standards in LC-MS/MS analysis.
Favorably, not just the synthesis of deuterium-labeled vitamin D metabolites is described, but also their application.
The synthesis of deuterium-labeled vitamin D metabolites, particularly labeled in the side chain (i.e. 6-fold labeling in C26/27-position or 3-fold labeling in 6,19,19-position is well established. Similar to the labeling in in 6,19,19-position the authors achieve a labeling by H/D-exchange, in contrast to the C26/27-labeling what is done by CD3-addition to a corresponding ester in the side chain. For LC-MS/MS applications usually a quantitative enrichment of labeled standard is essential. The H/D-exchange method has the disadvantage that the labeling may be insufficient and poorly reproducible. The authors admit, that the enrichment is just 93% (line 94), what I think might be too low for precise measurement of vitamin D metabolites by LC-MS/MS in biofluids. Additionally, the syntheses of enynes as A-ring synthons involve numerous synthesis steps and separation of stereoisomers is required. It’s a kind of pity, that, although starting with enantiomerically pure malic acid, the stereocenter isomerizes in the course of H/D-exchange. The synthesis of 16 requires almost 20 synthesis steps (!), although it is a relatively trivial compound. The yields are relatively poor in regard that just routine steps are applied (line 97: 22% from 10 to 11). Line 115/116: One expects two diastereomers, each as pair of enantiomers (i.e. 4 stereoisomers. How were enantiomers separated? By simple column chromatography?
Therefore, it becomes not really clear in the text what advantage a labeling in the A-ring comparing to labeling in the side chain may have, neither regarding the feasibility of synthesis nor the measurements in LC-MS/MS experiments. However, perhaps one may mention that the labeling in the A-ring may have an advantage in metabolism studies because the side chain is enzymatically degradated very easily, and derivatisation in the A-ring can also be favorable to install linkers for immunological assays for the development of specific antibodies that are able to discriminate metabolites varying in their side chain (Ref. 17).
I would suggest to better employ already labeled starting material for the synthesis of enynes instead, to achieve higher enrichment of the final standards and shorten their syntheses.
To my knowledge, the 23,26-lactone metabolites are less relevant because the corresponding enzyme does not occur in humans. By contrast, the corresponding vitamin D2 metabolites have to be considered as well and appear to me more clinically relevant than the 23,26 lactones.
Why were 25(R)24,25(OH)2D3 and its 1a-counterpart, along with 20(S)-metabolites not synthesized and tested, although these metabolites were mentioned in the abstract and introduction as relevant?
Minor corrections:
|
Line |
replace |
by |
comments |
|
15 |
disease |
diseases |
|
|
26 |
Measurement of blood levels |
Measurement of vitamin D metabolites in blood |
Or leave out |
|
33 |
kidney |
kidneys |
|
|
96 |
(diastereomeric mixture) |
(enantiomeric mixture) |
|
|
105 Scheme 1. |
Diastereomer mixture |
mixture of enantiomers |
|
|
106 |
a |
a |
|
|
113 |
b |
b |
|
|
186/187 |
labelled/labeled |
|
Spelling should be consistent throughout the text |
|
221 |
Amang, 1a,25di--- |
Among, 1a,25-di--- |
|
|
149 |
available |
suitable |
|
|
222 |
steroid |
Steroid |
|
|
124 |
C3a |
C3a |
|
|
231 |
C-11a |
C-11a |
|
Over all, the manuscript is well written with all care and presumably of significant interest for numerous readers/scientists working in the field. I recommend its publication with minor corrections.
Author Response
Dear Editor,
Thank you very much for your letter on April 4th 2022, with regarding our manuscript entitled “Synthesis of deuterium-labeled vitamin D metabolites as internal standards for LC-MS analysis” (Manuscript ID: molecules-1665794). We appreciate the reviewers’ efforts. In this letter, we described our responses and explanations to the reviewers’ comments. We also revised our manuscript according to the referee’s suggestions and comments.
We summarized our responses and revised points as follows, and we highlighted the revised parts with yellow in the revised manuscript.
Summary of Comments from the referees and our responses
Reviewer 1:
(1) The authors admit, that the enrichment is just 93% (line 94), what I think might be too low for precise measurement of vitamin D metabolites by LC-MS/MS in biofluids.
Our response
In this case, an enrichment level of 93% is enough to distinguish the non-labeled metabolites from the serum. We obtained the calibration curves based on the enrichment level, and we were able to measure the concentration of the metabolites correctly.
(2) Line 115/116: One expects two diastereomers, each as pair of enantiomers (i.e. 4 stereoisomers. How were enantiomers separated? By simple column chromatography?
Our response
Thank you for the valuable comment.
After coupling the A-ring with CD-ring, the resulting two diastereomers were separated by HPLC. We described the procedure in the experimental section, however as the reviewer pointed out, we added the procedure in the manuscript in the reference and footnotes “30”.
Revised
- The enantiomers at C3 were separated after coupling the CD rings by HPLC purification.
(3) Therefore, it becomes not really clear in the text what advantage a labeling in the A-ring comparing to labeling in the side chain may have, neither regarding the feasibility of synthesis nor the measurements in LC-MS/MS experiments. However, perhaps one may mention that the labeling in the A-ring may have an advantage in metabolism studies because the side chain is enzymatically degradated very easily, and derivatisation in the A-ring can also be favorable to install linkers for immunological assays for the development of specific antibodies that are able to discriminate metabolites varying in their side chain (Ref. 17).
Our response
Thank you for the valuable comment. According to the reviewer’s valuable suggestion, we added the following sentence in the manuscript.
Revised
line 95: In addition, labeling in the A-ring has an advantage in metabolism studies because the side chains of the D3 are well known to enzymatically metabolized easily.
(4) I would suggest to better employ already labeled starting material for the synthesis of enynes instead, to achieve higher enrichment of the final standards and shorten their syntheses.
Our response
Thank you for the valuable suggestion. We will try the synthesis in the next occasion.
(5) To my knowledge, the 23,26-lactone metabolites are less relevant because the corresponding enzyme does not occur in humans. By contrast, the corresponding vitamin D2 metabolites have to be considered as well and appear to me more clinically relevant than the 23,26 lactones.
Our response
Thank you for the valuable suggestion. We are now planning to synthesize the VD2 series according to the D3 series. We would like to report these results on another occasion.
Thank you again for the suggestion.
(6) Why were 25(R)24,25(OH)2D3 and its 1a-counterpart, along with 20(S)-metabolites not synthesized and tested, although these metabolites were mentioned in the abstract and introduction as relevant?
Our response
Thank you for the valuable comment. We are now trying to synthesize these compounds. After the synthesis, we will try to measure the concentration of these metabolites.
Minor corrections:
Our response
We appreciate these comments, and we apologize for these mistakes and typos in the manuscripts. As the reviewer pointed out, we revised all listed errors as follows.
line 16
Original
~the diagnosis of bone metabolism-related disease,
Revised
~the diagnosis of bone metabolism-related diseases,
line 27
Original
Measurement of blood levels
Revised
Measurement of vitamin D metabolites in blood
line 34
Original
~by CYP27B1-mediated oxidation at C1a in the kidney
Revised
~by CYP27B1-mediated oxidation at C1a in the kidneys
line 111
Original
The deuterium-labeled alcohol 10 (diastereomeric mixture) was~
Revised
The deuterium-labeled alcohol 10 (enantiomeric mixture) was~
Scheme 1.
Original
Diastereomer mixture
Revised
mixture of enantiomers
line 130
Original
~bearing a hydroxyl group at C1a was~
Revised
~bearing a hydroxyl group at C1a was~
line 135
Original
~and the undesired C1b diastereomer was~
Revised
~and the undesired C1b diastereomer was~
line 150
Original
~diastereomers at C3a were separated~
Revised
~diastereomers at C3a were separated~
line 180
Original
~the deuterium-labeled compounds are available as internal standards~
Revised
~the deuterium-labeled compounds are suitable as internal standards~
line 225
Original
We synthesized deuterium-labelled A-ring-d3 synthons~
Revised
We synthesized deuterium-labeled A-ring-d3 synthons~

Reviewer 2 Report
In the present manuscript, the authors reported the synthesis of deuterium-labelled A-ring-d3 synthons 13 and 16, and utilized them for the convergent synthesis of deuterium-labeled vitamin D3 derivatives 25(OH)D3 (2), 25(OH)D3-23, 26-lactone (6), and 1,25(OH)2D3-23, 26-lactone (7). These deuterium-labeled vitamin D3 metabolites were applied as internal standards for quantification of the metabolites in pooled human serum by LC MS/MS using the isotope dilution method. This is the first time quantification of 1,25(OH)2D3-23, 26-lac- tone (7) in human serum. The manuscript offer new deuterium-labeled synthesis methods and the information for quantification of vitamin D3 derivatives in human serum by LC MS/MS analysis. This paper was well organized and written.This study is of potential interest, I would like to recommend this manuscript to be published in Molecules after minor revision..
There are some unsatisfactory and should be revised in the manuscript..(1)For synthesized compounds, the state and color are missing. In support information, the HRMS of synthesized compounds should be provided. A graphical abstract is missing. (2) The compound configuration R-, L- , and Latin via, in situ, in vitro, in vivo should be italic. The first letter of every words of title or the only first letter of title in References be capitalized, shouled be unified. There were some mistake of type in manuscript, for example:vitamin D3 should be vitamin D3., CDCl3 should be CDCl3,la should be la , The author needs to check the article carefully.
Author Response
Dear Editor,
Thank you very much for your letter on April 4th 2022, with regarding our manuscript entitled “Synthesis of deuterium-labeled vitamin D metabolites as internal standards for LC-MS analysis” (Manuscript ID: molecules-1665794). We appreciate the reviewers’ efforts. In this letter, we described our responses and explanations to the reviewers’ comments. We also revised our manuscript according to the referee’s suggestions and comments.
We summarized our responses and revised points as follows, and we highlighted the revised parts with yellow in the revised manuscript.
Summary of Comments from the referees and our responses
Reviewer 2:
(1) For synthesized compounds, the state and color are missing. In support information, the HRMS of synthesized compounds should be provided.
Our response
Thank you for the comments.
We added the states and colors for the products in the experimental section. We also added the HRMS data in the SI (see p7 in the SI).
(2) A graphical abstract is missing.
Our response
Thank you for the comment. We thought we submitted the GA, may not be uploaded correctly. We re-uploaded the GA.
(3) The compound configuration R-, L- , and Latin via, in situ, in vitro, in vivo should be italic. The first letter of every words of title or the only first letter of title in References be capitalized, shouled be unified. There were some mistake of type in manuscript, for example:vitamin D3 should be vitamin D3., CDCl3 should be CDCl3,la should be la , The author needs to check the article carefully.
Our response
Thank you for the valuable comments. We fixed the problems pointed out by the reviewer described above.
